# Local Inflammatory Mediators Involved in Neuropathic Pain

**DOI:** 10.3390/ijms24097814

**Published:** 2023-04-25

**Authors:** Patricia García-Fernández, Colette Reinhold, Nurcan Üçeyler, Claudia Sommer

**Affiliations:** Department of Neurology, University Hospital of Würzburg, 97080 Würzburg, Germany; garcia_p@ukw.de (P.G.-F.);

**Keywords:** polyneuropathy, pain, inflammation, NF-κB, TNFα

## Abstract

Polyneuropathy (PNP) is a term to describe diseases of the peripheral nervous system, 50% of which present with neuropathic pain. In some types of PNP, pain is restricted to the skin distally in the leg, suggesting a local regulatory process leading to pain. In this study, we proposed a pro-inflammatory pathway mediated by NF-κB that might be involved in the development of pain in patients with painful PNP. To test this hypothesis, we have collected nerve and skin samples from patients with different etiologies and levels of pain. We performed RT-qPCR to analyze the gene expression of the proposed inflammatory pathway components in sural nerve and in distal and proximal skin samples. In sural nerve, we showed a correlation of TLR4 and TNFα to neuropathic pain, and an upregulation of TNFα in patients with severe pain. Patients with an inflammatory PNP also presented a lower expression of TRPV1 and SIRT1. In distal skin, we found a reduced expression of TLR4 and miR-146-5p, in comparison to proximal skin. Our findings thus support our hypothesis of local inflammatory processes involved in pain in PNP, and further show disturbed anti-inflammatory pathways involving TRPV1 and SIRT1 in inflammatory PNP.

## 1. Introduction

Polyneuropathy (PNP) is a term to describe diseases of the peripheral nervous system. This can include dysfunction of the motor, sensory, or autonomic system, leading to symptoms such as weakness, numbness, paresthesia, and pain [1]. While a wide range of underlying causes can result in PNP, up to 30% are idiopathic. About 50% of all PNP are associated with neuropathic pain [1,2,3]. Current treatments have shown little effect in the relief of neuropathic pain [4,5]. It is therefore imperative to find common pathophysiological pathways that might determine the severity of the disease and the development of pain, and guide the search towards effective therapies.

Many neuropathies are accompanied by inflammation, even if they do not belong to the classic immune mediated disorders [1,6]. Immune processes and mediators can be involved in neurodegeneration, -regeneration, and pain. For instance, an upregulation of pro-inflammatory cytokines, including tumor necrosis factor (TNFα) or interleukin (IL)-1β, may lead to pain, while anti-inflammatory cytokines such as IL-10 can have analgesic properties. A balance between these components seems to be key to maintain a beneficial equilibrium between neurodegeneration and -regeneration [7].

In PNP, several clinical studies point towards an enhanced systemic inflammation associated with the development of the neuropathy. These studies, nevertheless, have shown a high variability of results, suggesting that the analysis of local inflammatory components might be more valuable [8,9,10,11,12,13,14]. Supporting this are the increased number of macrophages and T-cells in the skin of patients with PNP, independent of its etiology [15], as well as elevated mRNA levels of TNFα, IL-1, IL-6, IL-8, or IL-10 [16,17]. Moreover, fibroblasts isolated from the skin of patients with neuropathies are able to secrete high levels of IL-6 and IL-8 [18].

This local inflammatory process might be triggered by the activation of toll-like receptor (TLR)-mediated pathways. In particular, stimulation of TLR4 results in the release of pro-inflammatory cytokines, such as TNFα, through the upregulation of an NF-κB-mediated cascade [19]. In turn, the NF-κB complex can be inhibited by the NAD-dependent deacetylase sirtuin 1 (SIRT1) via the deacetylation of its p65 subunit [20]. Activation of ion channels, such as the transient receptor potential cation channel 1 (TRPV1), trigger a strong influx of calcium, activating SIRT1 through a calcium-dependent protein kinase, and thus inhibiting the immune response [21,22]. On the other hand, activation of TRPV1 results in enhanced excitatory responses in nociceptors, promoting axonal degeneration and the release of pro-inflammatory mediators [23]. This dichotomy leads to the question of whether and how other factors in the local environment determine the actions of TRPV1.

Furthermore, small fragments of RNA, or microRNAs (miR), can regulate the expression of different inflammatory mediators. For instance, miR-146a-5p, miR-132-3p, and miR-155-5p are able to modulate the NF-κB mediated cascade and have been found altered in patients with neuropathic pain [24,25].

PNP symptoms are usually length-dependent and more severe in the extremities [26]. A local study of the inflammatory environment allows the comparison between painful regions of skin, located distally in the leg, and painless regions at the proximal leg. Therefore, this is a good model to study the local role of an immune imbalance within the same patient.

Here, we hypothesized that a local pro-inflammatory environment is involved in the severity of the disease and the development of neuropathic pain in PNP. To test our hypothesis, we studied the gene expression of TLR4, TRPV1, TNFα, SIRT1, miR-146a-5p, miR-132-3p, and miR-155-5p, in sural nerve, and in proximal and distal skin from patients with PNP. We correlated their expression to different neuropathy scores, disease severity, and levels of pain. Our results highlight the importance of the immune balance in the development of neuropathic symptoms.

## 2. Results

### 2.1. Clinical Characteristics of Patient Cohort

Diagnostic subgroups and clinical characteristics are summarized in Table 1. Sixty-six patients with PNP that presented for diagnostic work-up and had a sural nerve biopsy were included in the study (median age 65 years, range 28–87). The median disease duration was 3 years (range 0.01–27 years). Overall, 40 patients were diagnosed with an inflammatory neuropathy including non-systemic vasculitis (16 patients), monoclonal IgM gammopathy (5 patients), chronic inflammatory demyelinating polyneuropathy (CIDP) (6 patients), and other inflammatory neuropathies [Guillain-Barré syndrome (GBS, 1 patient), ganglionitis (1 patient), multiple myeloma (3 patients), and other (8 patients)]. In total, 26 patients were categorized as non-inflammatory, including idiopathic neuropathy (14 patients), hereditary neuropathy (5 patients), diabetic neuropathy (3 patients), vitamin B12 deficiency (2 patients), motor neuron disease (MND, 1 patient) and AL-amyloidosis (1 patient).

Patients were further divided according to their levels of pain on the numerical rating scale (NRS): No pain (NRS = 0; 17 patients), mild pain (1 ≤ NRS ≤ 3; 22 patients), and severe pain (NRS ≥ 4; 27 patients). At the time of inclusion, 10 patients had been treated for their neuropathy with intravenous immunoglobulin therapy (IVIG) within the last 6 months, while 56 had not received treatment.

While all patients received a sural nerve biopsy, in 33 out of 66 patients, a biopsy of skin from the lower leg (distal skin) and the upper thigh (proximal skin) was also collected. The analysis of the intraepidermal nerve fiber density (IENFD) showed a median of 2.1 fibers/mm (range 0.0–12.3) in the distal and 7.0 fibers/mm (0.0–18.0) in the proximal skin.

### 2.2. Gene Expression of Inflammatory Markers in Sural Nerve

In the 66 PNP patients that comprise the study cohort, we analyzed the gene expression of the receptors TLR4 and TRPV1, the deacetylase SIRT1, the pro-inflammatory cytokine TNFα, and the microRNAs miR-146a-5p, miR-132-3p, and miR-155-5p, in sural nerve biopsies. To control for the effect of the IVIG treatment, we analyzed this subgroup separately. While no difference was found for TLR4, TNFα, SIRT1, miR-132-3p, or miR-146-5p, TRPV1 was upregulated in patients with IVIG treatment in comparison to those without treatment (*p* < 0.001). Patients with this treatment also presented a downregulation of miR-155-5p (*p* < 0.01). The effects of IVIG treatment on each of these markers were confirmed with the 95% confidence intervals (Appendix B, Figure A1A). After this discovery, the analyzes of TLR4, TNFα, SIRT1, miR-132-3p, and miR-146-5p were performed in the full cohort of 66 PNP patients, while TRPV1 and miR-155-5p were investigated in the 55 patients without prior treatment.

Comparisons between pain subgroups (no, mild, and severe pain) and between patients with an inflammatory and non-inflammatory neuropathy showed no differences in the expression of TLR4 (Figure 1A), and the microRNAs miR-146a-5p, miR-132-3p, and miR-155-5p (Figure 1E). On the other hand, patients with severe pain presented an upregulated expression of TNFα in comparison to those with no (*p* < 0.05) or mild pain (*p* < 0.01) (Figure 1B). Although this upregulation was limited to patients with an inflammatory neuropathy, the expression of TNFα correlated with the severity of PNP in the entire cohort, with its highest values in patients with an inflammatory neuropathy and severe pain and its lowest values in patients with a non-inflammatory PNP and no pain (*p* < 0.001) (Figure 1F). Furthermore, patients with an inflammatory PNP also presented a downregulation of TRPV1 (*p* < 0.05) (Figure 1C) and SIRT1 (*p* < 0.05) (Figure 1D) in comparison to those with a non-inflammatory one.

### 2.3. Gene Expression of Pro- and Anti-Inflammatory Markers in Skin

Thirty-three PNP patients out of the sixty-six included in the study received a skin punch biopsy at a proximal (upper thigh) and distal (lower leg) region, where the same inflammatory markers were analyzed. In this case, patients with IVIG treatment did not show any differential expression of the studied components in comparison to those without treatment (Appendix B, Figure A1B,C). Therefore, the following analyzes were performed in the entire 33-patient cohort.

#### 2.3.1. Inflammatory Markers in Neuropathic Pain

In distal skin, no differences in the expression of the studied inflammatory markers were found between pain subgroups (no, mild, and severe pain). In proximal skin, an increased expression of miR-146-5p was observed in patients with severe pain, in comparison to those with mild pain (*p* < 0.05) (Figure 2E), while no differences were discovered for TLR4 (Figure 2A), TRPV1 (Figure 2C), TNFα (Figure 2B), SIRT1 (Figure 2D), miR-132-3p, and miR-155-5p (Figure 2E).

Paired analyzes between distal and proximal skin allowed the study of these components between painful (distal) and painless (proximal) regions within the same patient. Our results showed a downregulation in the expression of TLR4 (*p* < 0.05) and miR-146-5p (*p* < 0.05) in distal skin in comparison to proximal, specifically in those patients with mild or severe pain (Figure 2A,E).

#### 2.3.2. Inflammatory Markers in Inflammatory and Non-Inflammatory Neuropathies

Comparison between patients with an inflammatory and non-inflammatory neuropathy did not show any differences in the expression of the studied inflammatory markers (Figure 3). Paired analyzes between distal and proximal skin showed a lower expression of miR-146-5p in distal skin in comparison to proximal (*p* < 0.01) (Figure 3E), in all PNP patients, and of TLR4 specifically in those patients with an inflammatory neuropathy (*p* < 0.01) (Figure 3A).

#### 2.3.3. Inflammatory Markers and Intraepidermal Nerve Fiber Density (IENFD)

In each skin biopsy, the IENFD can be used to determine the level of degeneration of nerve fibers in the epidermis and dermis. Correlation analyzes between the expression of the studied inflammatory components and the IENFD can be performed to determine their role in nerve degeneration. Our results showed that while in distal skin no correlation was found between the markers and IENFD (Figure 4A), in proximal skin miR-155-5p was negatively related to IENFD (*p* < 0.05) (Figure 4B).

Moreover, the correlation between distal and proximal expression of each inflammatory component was analyzed to determine the systemic aspect of their role in PNP. Our analysis showed that the expression of TNFα (*p* < 0.05) and SIRT1 (*p* < 0.0001) correlated between both regions, suggesting a systemic modulation of their expression.

### 2.4. Correlation of Inflammatory Markers with the Severity of PNP

All PNP patients included in the study were examined to determine the severity of their neuropathy with standardized scales, including the modified Toronto clinical neuropathy score (mTCNS), the overall disability sum score (ODSS), and the Medical Research Council (MRC)-sumscore, while pain was specifically evaluated with NRS. In order to determine the involvement of the inflammatory markers in the development of the neuropathy and of neuropathic pain, these neuropathy scales were correlated with the analyzed markers in sural nerve and distal and proximal skin. Our analysis of sural nerve showed that NRS correlated positively with both TLR4 (*p* < 0.05) and TNFα (*p* < 0.001). Furthermore, miR-132-3p expression correlated with TCNS (*p* < 0.01) and MRC (*p* < 0.05), and negatively with NRS (*p* < 0.01) (Figure 5A).

In distal skin, our analysis showed that ODSS correlated with miR-132-3p (*p* < 0.05) (Figure 5B), while in proximal skin NRS correlated with miR-146-5p (*p* < 0.05) and miR-155-5p (*p* < 0.05) (Figure 5C). None of the other markers showed correlations with neuropathy scores.

## 3. Discussion

In this study, we analyzed the gene expression of different inflammatory components of the NF-κB pathway in sural nerve and proximal and distal skin from patients with PNP. With this analysis, we aimed to elucidate whether a local pro-inflammatory environment mediated by NF-κB may determine the severity of the disease and the presence of neuropathic pain.

The NF-κB-mediated pathway can be activated by the stimulation of TLR4 with a damage-associated molecular pattern (DAMP), resulting in the release of pro-inflammatory cytokines such as TNFα [19,27,28,29]. Our results in sural nerve showed that the expression of TLR4 and TNFα correlated with each other and with the NRS, indicating a connection to the development of neuropathic pain. Furthermore, TNFα expression levels correlated positively with the severity of the neuropathy and were particularly elevated in the presence of severe pain in patients with inflammatory PNP. It is likely that the amount of endogenous TLR4 ligands, like high mobility group box1 (HMGB1), is increased in painful neuropathy, binding to TLR4 and leading to the activation of the NF-κB pro-inflammatory cascade and the upregulation of TNFα [30] (Figure 6).

Since TLRs are expressed in most cell types, fibroblasts and Schwann cells that are supporting the nerve fibers might be involved in the upregulation of this cascade. However, fibroblasts and Schwann cells express low levels of TLR4 (20.5 and 0.8 normalized protein-coding transcripts per million [nTPM], respectively) and TNFα (1.5 and 12.7nTPM, respectively). Myeloid cells on the other hand, such as monocytes, macrophages, or T-cells, express higher levels of TLR4 (118.1 nTPM, 89.3 nTPM, and 0.9 nTPM, respectively) and TNFα (54 nTPM, 116.1 nTPM, and 94.6 nTPM, respectively) in order to respond against stimuli [28,31,32]. Our data thus suggest that in sural nerve of patients with severe PNP, resident or infiltrated myeloid cells may produce TNFα via activation of TLR4 (Figure 6). 

These results also confirm the previous literature that describes TNFα as closely related to many neuropathies and, in particular, to neuropathic pain. Not only has TNFα been reported upregulated in the peripheral and central nervous system in animal pain models, but its administration has also reproduced pain hypersensitivity. The mechanisms in which TNFα may induce neuropathic pain are plenty and very variable. One of the proposed mechanisms is through the regulation of cation channels, leading to sensitization of primary afferents, neuronal excitability, and repetitive firing. Another suggested mechanism would be through the activation of NF-κB, p38 Mitogen-activated protein kinase (MAPK) and c-Jun N-terminal kinases (JNK) pathways via TNF receptor 1 (TNFR1), promoting pro-inflammatory loops and initiating apoptotic processes that contribute to nerve degeneration and pain [28,33,34].

In addition, we found a close relation between TRPV1 and SIRT1, both being downregulated in sural nerve from patients with inflammatory PNP. SIRT1 is able to deacetylate NF-κB, and its downregulation would imply an upregulation of the NF-κB pathway, explaining the high expression of TNFα [20]. While there are a few candidates that may inactivate or inhibit the activity of SIRT1, little is known about the regulation of its expression [35,36,37]. One possibility is through the formation of a silencing complex between a microRNA and the SIRT1 transcript, promoting its destabilization and/or translational repression. In particular, two of the studied microRNAs, miR-132-3p and miR-155-5p, are able to bind and silence the expression of SIRT1. While the expression levels of miR-132-3p and miR-155-5p correlated strongly with each other, they did not differ between patients with inflammatory and non-inflammatory neuropathy, thus suggesting that other microRNAs might be involved in this regulation, including miR-138-5p, miR-9-5p, or miR-22-3p [38,39]. Furthermore, a relation between TRPV1 and SIRT1 has been described before. Therefore, we cannot exclude the possibility of a downregulation of TRPV1 leading to a lower expression of SIRT1, or vice versa [21,40,41,42,43,44]. Our data thus suggests that a downregulation of SIRT1 via TRPV1, or potentially other microRNAs than miR-132-3p and miR-155-5p, might be involved in the development of an inflammatory PNP.

Since PNP symptoms are usually length dependent and more severe in the extremities, the analysis of regions of skin distally located can provide clarity on the inflammatory components associated with more severe and painful neuropathic symptoms [26]. Our results did not show any differences in the expression of TNFα, TRPV1, SIRT1, miR-132-3p, or miR-155-5p between distal and proximal regions of skin from PNP patients. On the other hand, the expression of TLR4 and miR-146a-5p was found downregulated in distal versus proximal skin regions, which replicates previous findings from our group [25]. MiR-146a-5p is a microRNA that can form a silencing complex with the interleukin-1 receptor-associated kinase 1 (IRAK1) and TNF receptor-associated factor 6 (TRAF6) transcripts, components of the NF-κB pathway. Activation of the NF-κB cascade can lead to the upregulation of miR-146a-5p, in order to repress components of the same pathway and control the inflammation [38,39,45,46,47]. The low expression of miR-146a-5p might be due to a downregulation of TLR4, and thus of the NF-κB pathway. This repression of TLR4 might be the result of a regulatory loop being activated upon pro-inflammatory mechanisms taking place in distal skin and being associated with severe neuropathic symptoms.

Furthermore, we found that the expression of SIRT1, as well as TNFα, correlated positively between distal and proximal regions of skin, suggesting a systemic regulation of these markers, and highlighting their involvement in systemic inflammation. Moreover, the current literature has reported involvement of SIRT1 in systemic processes such as ageing, metabolism, diet, glucose tolerance, or immune disorders [48,49,50,51,52,53,54,55]. Although we did not find correlations between SIRT1 and age, we cannot discard other patient-inherent processes altering its expression.

While our study includes some of the main inflammatory components of the NF-κB pathway, a broader analysis might help elucidate other pathways and mediators that might be involved in PNP. Furthermore, though sural nerve cannot be obtained from healthy individuals, skin biopsies could be performed to include a control group. This may clarify the role of the studied inflammatory components in the onset and worsening of the neuropathy. In addition, the analysis of the protein levels of these inflammatory mediators might shed light on the post-translational regulations that they might encounter and the inter-modulatory activities affecting the NF-κB pathway. Lastly, our results are limited by the well characterized but low number of recruited patients and the variance in etiologies. A bigger cohort might yield clearer results.

At the moment, first-line treatments of neuropathic pain are based on the use of antidepressant and antiepileptic drugs, while opioids are recommended as second- and third-line treatment. These therapies are ineffective in many patients or inappropriate for specific painful neuropathies. Many current efforts are being made into identifying new targets to develop novel pharmaceutical agents [4,5]. Our study highlights the role of the NF-κB pathway in PNP and uncovers new targets that could be used towards finding better treatments against neuropathic pain. Furthermore, current therapeutic options targeting inflammatory mediators, such as the cyclooxygenase-2 [56], reducing the levels of TNFα [57,58], or inhibiting the activation of immune cells via the fractalkine receptor [59], are being explored.

We conclude that stimulation of TLR4 by endogenous ligands in large nerve trunks may result in the upregulation of TNFα, via the activation of the NF-κB pathway, leading to the development of severe neuropathic pain in patients with PNP. Furthermore, we propose that the activation of the NF-κB pathway in large nerve fibers, through the downregulation of SIRT1 and TRPV1, might be involved in the pathophysiology of inflammatory neuropathies. From our results in skin, we cannot resolve the involvement of the NF-κB pathway in the degeneration of nerve fiber endings and the development of localized neuropathic pain. Further analysis of this and other inflammatory pathways may help elucidate the role of inflammation in the skin of patients with PNP. We believe that the study of the NF-κB pathway in large nerve fibers can be an additional indicator of the severity of PNP and improve patient stratification, while a follow-up study may promote the discovery of predictive biomarkers.

## 4. Materials and Methods

### 4.1. Patient Recruitment and Sample Collection

Between 2017 and 2018, 66 patients with PNP that came to the Department of Neurology, University of Würzburg, Germany for diagnostic work-up, were recruited for this study. Patients were informed about the aim in which their tissue was going to be used and gave written informed consent prior to inclusion. This project was approved by the Würzburg Medical Faculty Ethics Committee (# 230/15).

Patients were diagnosed based on neurological examinations, laboratory studies, and nerve conduction examinations. The severity of the neuropathy was determined using standardized scales, including the modified Toronto clinical neuropathy score (mTCNS), the overall disability sum score (ODSS), and the Medical Research Council-sumscore (MRC-sumscore). Pain was evaluated by a numerical rating scale (NRS) from 0 (no pain) to 10 (worst pain).

From each patient, a biopsy of sural nerve was collected and embedded in RNA-later over night at 4 °C for gene expression analysis. The RNA-later was removed the next day and the samples were preserved at −80 °C until further analysis. From 33 out of 66 patients, a biopsy of skin from the lower leg and the upper thigh was collected following the same protocol.

### 4.2. Gene Expression Analysis

#### 4.2.1. RNA Isolation

Purification of total RNA including miRNA was performed using the miRNeasy mini kit (Qiagen, Hilden, Germany), following the manufacturer’s protocol. In summary, samples were homogenized with 700 µL of Qiazol lysis reagent in a homogenizer disperser. After incubation with 140 µL of chloroform, samples were centrifuged for 15 min at 12,000× *g* at 4 °C in order to separate the aqueous phase (containing the RNA), the organic phase (containing proteins and lipids), and the interphase formed by DNA. The upper phase was collected in a column tube and RNA was cleansed through several centrifugation steps, upon addition of ethanol (EtOH), RWT buffer, and RPE buffer, while discarding the supernatant in every step. Lastly, RNA was retrieved by pipetting 33 µL of RNase-free water onto the membrane of the column and centrifuging the tubes for 1 min at 8000× *g*. Samples were stored at −80 °C after the RNA quality and quantity was assessed with a NanoDrop™ One (Thermo Fisher Scientific, Waltham, MA, USA).

#### 4.2.2. cDNA Synthesis

The cDNA synthesis from mRNA was carried out with TaqMan Reverse Transcription reagents (Thermo Fisher Scientific, Waltham, MA, USA). First, 250 ng mRNA of each sample was pre-incubated with 5 µL random hexamer at 85 °C for 3 min. Next, 10 μL 10× PCR buffer, 22 μL MgCl_2_, 20 μL deoxyribonucleoside triphosphate, 6.25 μL multiscribe reverse transcriptase, and 2 μL RNase inhibitor was added per sample. Lastly, reaction was performed under these conditions: annealing (25 °C, 10 min), reverse transcription (48 °C, 60 min), and enzyme inactivation (95 °C, 5 min).

For miRNA, reverse transcription was performed with the miRCURY LNA RT Kit (Qiagen, Hilden, Germany). For each sample, 10 ng of RNA was mixed with 2 μL of 5x reaction buffer, 5 μL of nuclease free water, and 1 μL of enzyme mix. Reaction was performed using the following program: reverse transcription (42 °C, 60 min) and enzyme deactivation (95 °C, 5 min).

Reactions were carried out on a PRISM 7700 Cycler (Applied Biosystems, Waltham, MA, USA) and transcribed cDNA was stored at −20 °C until further analysis.

#### 4.2.3. RT-qPCR

Real-time qPCR of mRNA and miRNA targets was performed to analyze gene expression using the StepOnePlus Real-Time PCR System (Thermo Fisher Scientific, Waltham, MA, USA). For mRNA, RT-qPCR was carried out with TaqMan qRT-PCR reagents (all Thermo Fisher Scientific, Waltham, MA, USA) and pre-designed assays. For target normalization, experience from our group showed that the ribosomal protein L13a (RPL13A) could act as a good housekeeping gene. Based on its comparability and standard deviation across groups and samples, our tests showed that RPL13A could be selected as suitable endogenous control. For each reaction, 3.5 µL cDNA was mixed with 0.5 µL nuclease free water, 5 µL Fast Advanced Mastermix, and 0.5 µL TBP primer and 0.5 µL target primer (see list of primers in Table 1).

For miRNA, the miRCURY LNA SYBR Green PCR Kit (Qiagen, Hilden, Germany) and pre-designed miRCURY LNA miR PCR assays (Qiagen, Hilden, Germany) were used. Based on previous experience from our group [11,12], the housekeeping genes 5S, SNORD44, SNORD48, and U6 were tested in nerve and skin samples as endogenous controls. Due to differences found between skin and nerve, different endogenous controls were selected for each sample type. Our tests showed that 5S and U6 were stable across groups in skin samples, while 5S and SNORD44 were more suitable for nerve samples. Each miRNA was run adding 5 µL 2× miRCURY SYBR Green Master Mix with 1 µL ROX per 50 µL and 1 µL primer (see list of primers in Table 2) to 4 µL of 1:80 diluted cDNA.

Each mRNA and miRNA was amplified in triplicates and relative quantitation (RQ) values were obtained by the StepOnePlus™ Software v2.3 (Thermo Fisher Scientific, Waltham, MA, USA) using interplate calibrators through the ∆∆Ct method.

### 4.3. Statistical Analysis and Visualization

Statistical analysis was performed in SPSS 27 (IBM, Armonk, NY, USA), where the Shapiro–Wilk test was used to determine the normal distribution of the data. For parametric data, a *t*-test was used for comparison between two groups, and One-Way ANOVA for three groups. In non-parametric data, the Mann–Whitney U Test was applied for comparison of two groups, and Kruskal–Wallis for three groups. The Spearman test was used for correlations between data groups. Comparisons between distal and proximal skin were analyzed through dependent-samples Wilcoxon signed-rank test. Data results were plotted in GraphPad Prism 9 (GraphPad Software, Inc., La Jolla, CA, USA) for visualization.

## Figures and Tables

**Figure 1 ijms-24-07814-f001:**
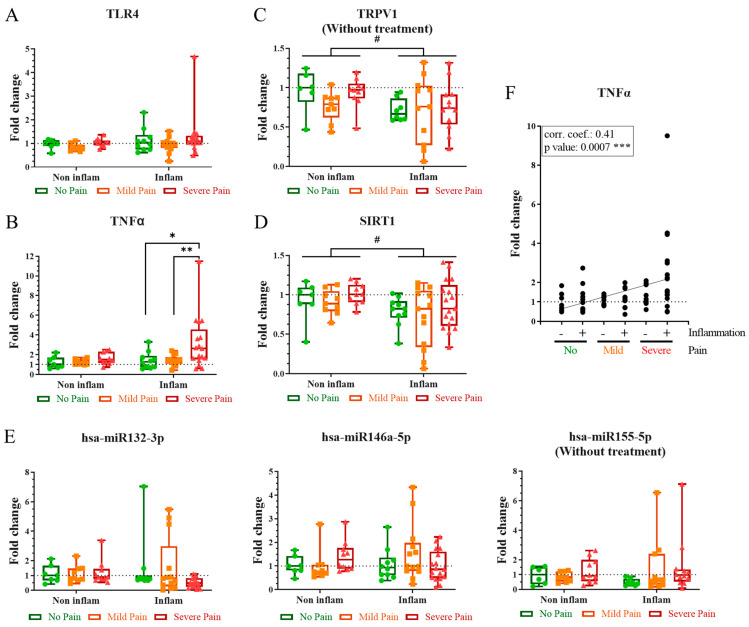
Inflammatory markers involved in the severity of PNP in sural nerve biopsies. Gene expression of TLR4 (**A**), TNFα (**B**), TRPV1 (**C**), SIRT1 (**D**), and the microRNAs miR132-3p, miR146a-5p, and miR155-5p (**E**) in sural nerve from patients with no (green), mild (orange), or severe (red) pain, and divided according to their inflammatory or non-inflammatory neuropathy. The number of patients included in each subgroup can be found in Table A1. Results are normalized to patients with no pain and a non-inflammatory neuropathy. Kruskal–Wallis tests were performed between pain subgroups (*) and Mann–Whitney tests between inflammatory subgroups (#). (**F**) Spearman correlation between the expression of TNFα and the severity of the neuropathy. Corr. coef., correlation coefficient; Inflam, inflammatory PNP; Non inflam, non-inflammatory PNP. */#, *p* < 0.05; **, *p* < 0.01; ***, *p* < 0.001.

**Figure 2 ijms-24-07814-f002:**
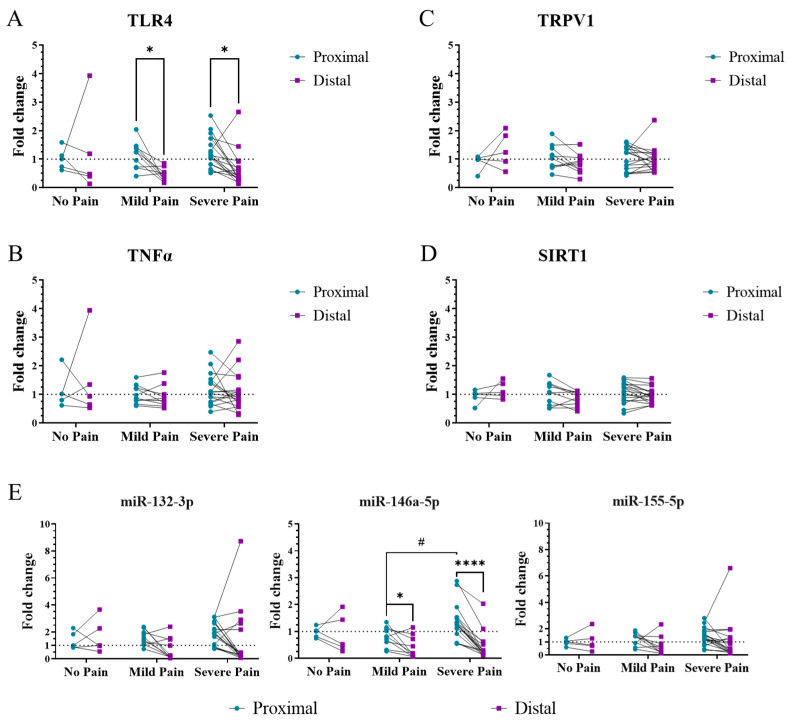
TLR4 and miR-146a-5p differ between distal and proximal skin regions in PNP patients with mild and severe pain. Gene expression of TLR4 (**A**), TNFα (**B**), TRPV1 (**C**), SIRT1 (**D**), and the microRNAs miR-132-3p, miR-146-5p, and miR-155-5p (**E**) in distal (purple) and proximal (turquoise) skin from patients with no (*n* = 5), mild (*n* = 10), or severe (*n* = 18) pain. Results are normalized to the proximal skin of patients with no pain. Kruskal–Wallis tests were performed between pain subgroups (#) and related-samples Wilcoxon Signed Rank tests between distal and proximal skin (*). #/*, *p* < 0.05; ****, *p* < 0.0001.

**Figure 3 ijms-24-07814-f003:**
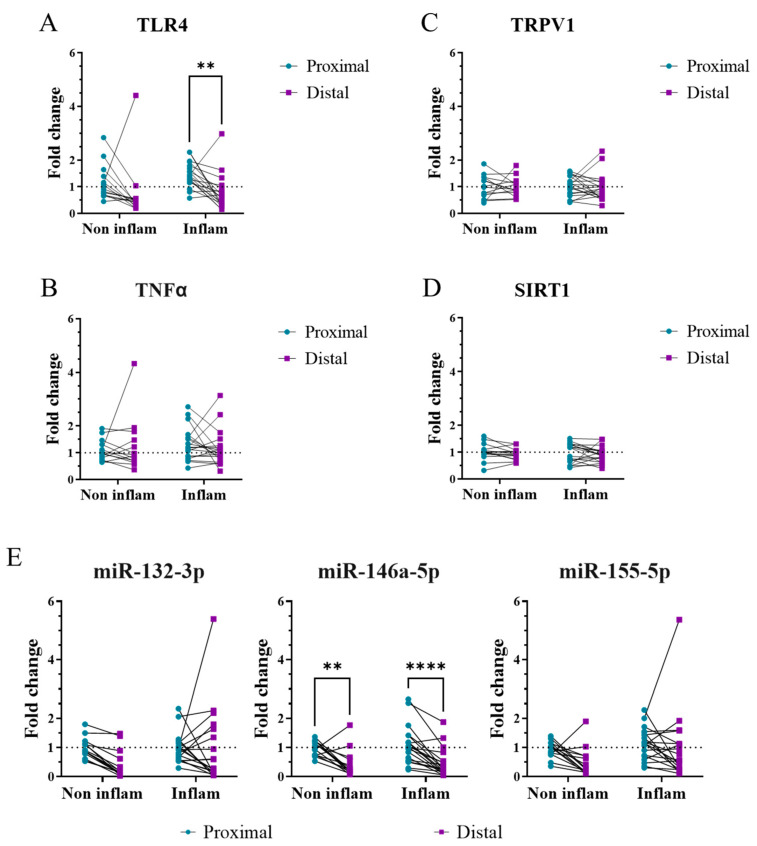
TLR4 and miR-146a-5p differ between distal and proximal skin regions in patients with inflammatory PNP. Gene expression of TLR4 (**A**), TNFα (**B**), TRPV1 (**C**), SIRT1 (**D**), and the microRNAs miR-132-3p, miR-146-5p, and miR-155-5p (**E**) in distal (purple) and proximal (turquoise) skin from patients with an inflammatory (n = 19) and non-inflammatory (n = 14) neuropathy. Results are normalized to the proximal skin of patients with a non-inflammatory neuropathy. Mann–Whitney tests were performed between inflammatory subgroups (not significant) and related-samples Wilcoxon Signed Rank tests between distal and proximal skin (*). Inflam, inflammatory PNP; Non inflam, non-inflammatory PNP. **, *p* < 0.01; ****, *p* < 0.0001.

**Figure 4 ijms-24-07814-f004:**
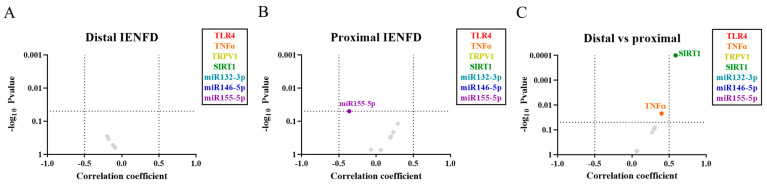
Most inflammatory markers do not correlate to the IENFD in distal and proximal skin regions of PNP patients. Volcano plot of Spearman correlations between IENFD and the gene expression of the inflammatory markers (in legend) in distal (**A**) and proximal (**B**) skin from patients with PNP. (**C**) Volcano plot of Spearman correlations between the gene expression of each inflammatory marker in distal versus proximal skin regions. Correlations with a *p* < 0.05 are colored, while those not significant (*p* ≥ 0.05) are marked in grey. IENFD, intraepidermal nerve fiber density.

**Figure 5 ijms-24-07814-f005:**
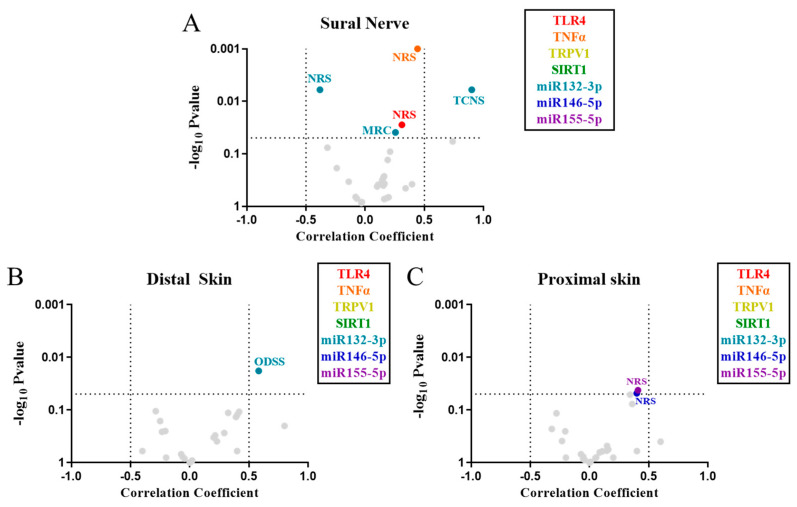
Inflammatory markers correlate with the severity of PNP. Volcano plot of Spearman correlations between the gene expression of the inflammatory markers (in legend) and four neuropathy scores (MRC, NRS, ODSS, and TCNS), in sural nerve (**A**) and distal (**B**) and proximal (**C**) skin. Correlations with a *p* < 0.05 are colored, while those not significant (*p* ≥ 0.05) are marked in grey. MRC, Medical Research Council-sumscore; NRS, numerical rating scale; ODSS, overall disability sum score; TCNS, modified Toronto clinical neuropathy score.

**Figure 6 ijms-24-07814-f006:**
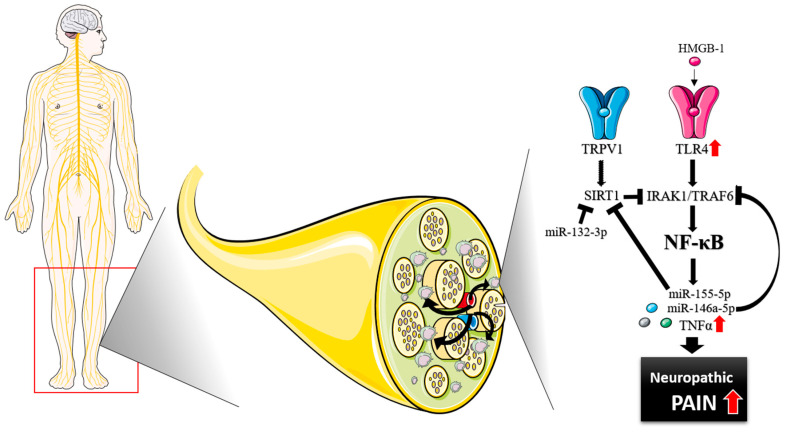
Graphical abstract of the studied inflammatory pathway and the results regarding TLR4/NF-κB signaling and TNFα production in neuropathic pain in sural nerve. Graphical images were incorporated from Smart Servier Medical Art, https://smart.servier.com/, accessed on 12 April 2023, under the Creative common Attribution 3.0 Unported Licence.

**Table 1 ijms-24-07814-t001:** Diagnostic subgroups and clinical characteristics of patient cohort.

Item	Number (% of Group)
M, F (*N*)	40, 26
Median age (range)	65 years (28–87)
Median disease duration (range in years)	3 years (0.01–27)
Diagnostic subgroups (*N* and % of entire group):	
Vasculitic neuropathy	16 (24.2%)
Idiopathic neuropathy	14 (21.2%)
Chronic inflammatory demyelinating polyneuropathy (CIDP)	6 (9.1%)
Monoclonal IgM gammopathy	5 (7.6%)
Hereditary neuropathy	5 (7.6%)
Diabetic neuropathy	3 (4.5%)
Vitamin B12 deficiency	2 (3.0%)
Motor neuron disease (MND)	1 (1.5%)
AL-amyloidosis	1 (1.5%)
Other inflammatory neuropathies [Guillain-Barré syndrome (GBS), ganglionitis, multiple myeloma, other]	13 (19.7%)
Analysis subgroups (*N*):	
Inflammatory, non-inflammatory neuropathy	40, 26
No pain, mild pain, severe pain	17, 22, 27
Patients with prior IVIG treatment, without treatment (*N*)	10, 56
Biopsies (*N*):	
Sural nerve biopsy	66
Distal and proximal skin biopsy	33
Intraepidermal nerve fiber density (IENFD)	
Median distal IENFD (fibers/mm) (range)	2.1 (0.0–12.3)
Median proximal IENFD (fibers/mm) (range)	7.0 (0.0–18.0)

**Table 2 ijms-24-07814-t002:** List of primer assays.

Taqman Primer *	Assay Number
TLR4	Hs00152939_m1
TNFα	Hs00174128_m1
TRPV1	Hs00218912_m1
SIRT1	Hs01009006_m1
RPL13A	Hs04194366_g1
**SYBR Green Primer ^#^**	**Assay Number**
hsa-miR-132-3p	YP00206035
hsa-miR-146a-5p	YP00204688
hsa-miR-155-5p	YP00204308
SNORD44	YP00203902
U6	YP02119464
5S rRNA	YP00203906

* Taqman Primers were purchased from Thermo Fisher Scientific, Waltham, MA, USA. ^#^ SYBR Green Primers were purchased from Qiagen, Hilden, Germany. Abbreviations: TLR4, toll-like receptor 4, TNFα, tumor necrosis factor α; TRPV1, transient receptor potential cation channel subfamily V member 1; SIRT1, sirtuin 1; RPL13A, Ribosomal protein L13a; miR, microRNA; SNORD, Small nucleolar RNAs C/D box; rRNA, ribosomalRNA.

## Data Availability

Data is contained within the article or Appendix A.

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
