# Peer review of "Local Inflammatory Mediators Involved in Neuropathic Pain"

_ijms, 2023, doi:10.3390/ijms24097814_

Round 1

Reviewer 1 Report

The manuscript by García-Fernández et al. “Local inflammatory mediators involved in neuropathic pain” describes the potential NF-κB-mediated gene expression using RT-qPCR to analyze the gene expressions of TLR4, TRPV1, SIRT1, TNFα, and microRNAs (miR) in the sural nerve and the skin from patients with various painful polyneuropathy.

The issue of the NF-κB-mediated gene expression for the pain-related receptors/ion channels, signaling proteins, and pro-inflammatory mediators that maybe associated with peripheral sensitization in clinical polyneuropathy is a very interesting subject in light of recent reports of its efficiency in pain treatment as well as from mechanistic points of view.

Main concerns

1.     By sural nerve biopsies, the authors illustrated the TNFα gene expression (Fig. 1B), not the TLR4 gene expression (Fig. 1A), was correlated with the severe pain in inflammatory neuropathy. In line 235, however, their results in the sural nerve showed that the TLR4 and TNFα gene expressions were correlated with each other (data not shown), indicating a connection to the development of neuropathic pain. I am confused about two statements in your manuscripts. After all, the TLR4 gene expression showed no differences in the sural nerve between the inflammatory and non-inflammatory neuropathy.

2.     I am agreed that TLR4 activation leads to the production and release of TNFα through the NF-κB-mediated cascade (Han et al., 2021; Xing et al., 2018). However, the authors suggested resident or infiltrated myeloid cells, not fibroblasts or Schwann cells, might contribute to the neuropathic pain in sural nerve. Which type of myeloid cells – mononuclear (monocytes, macrophages, and dendritic cells) and polymorphonuclear (neutrophils, eosinophils, basophils, and mast cells) is the most important for TLR4 activation and TNFα production? Could you provide the morphological evidence from the sural nerve biopsies?

3.     The down-regulated TLR4 gene expressions (Fig. 2A), not the TNFα gene expressions (Fig. 2B), were showed in both the distal and proximal skin from those patients with mild or severe pain. In inflammatory neuropathy, similar gene expressions of TLR4 and TNFα were observed between the distal and proximal skin (Fig. 3A, B). According to your results in the skin, how to explain the role of TLR4/NF-κB signaling and TNFα production in neuropathic pain? Whether myeloid cells also mediate this process in the skin for inducing neuropathic pain?

4.     It is known that TRPV1 activates SIRT1 for inhibiting TNF-α production by blocking NF-κB-mediated gene expression (Chen et al., 2020). Due to the results in the sural nerve, the down-regulation of TRPV1 and SIRT1 gene expressions was observed in inflammatory neuropathy (Fig. 1C, D). However, these results in the skin did not show any differences in TRPV1 and SIRT1 gene expressions in inflammatory neuropathy (Fig. 3C, D). Upon these results, whether the role TRPV1/SIRT1 signaling is not activated for attenuating the inflammatory pain by the modulation of NF-κB-mediated gene expression in the skin?

5.     What’s more, the authors stated that miR-146a-5p gene expression, not miR-132-3p and miR-121 155-5p gene expression, showed a severe decrease in distal skin in patients with mild and severe pain (Fig. 2E), and even exhibited in the non-inflammatory and inflammatory neuropathy (Fig. 3E). As we have known, miR-146a-5p focuses on its role in regulating NF-κB-mediated gene expression in the various pathologies by inhibiting IRAK1 and TRAF6 in the TLR signaling (Paterson and Kriegel, 2017). It seems reasonable in your results; however, I do not know which type of myeloid cells has its potential role in the neuropathic pain through a down-regulation of miR-146a-5p gene expression in the skin? Importantly, in proximal skin, an increased miR-146-5p gene expression was observed in patients with severe pain, in comparison to those with mild pain (Fig. 2E). How do the authors explain this converse finding?

6.     In line 193, our analysis showed that the expression of TNFα (p < 0.05) and SIRT1 (p < 0.0001) correlated between both regions, suggesting a systemic modulation of their expression. I do not understand what is the systemic modulation? Additionally, whether the significant difference of SIRT1 (p < 0.0001) gene expression in the distal versus proximal skin (Fig. 4C) is analyzed from the values of Fig 2D or 3D?

7.    About the four neuropathy scores (MRC, NRS, ODSS and TCNS) in Fig. 5, their analysis of the sural nerve, not the skin, showed that NRS correlated positively with both TLR4 (p < 0.05) and TNFα (p < 0.001). It seems to be important to confirm the role of TLR4/NF-κB signaling and TNFα production in sural nerve that bringing up the neuropathic pain after clinical polyneuropathy. More importantly, I suggest the authors should be concerned about the critical role of myeloid cells in neuropathic pain and privide the evidence in the sural nerve by in situ hybridization and immunohistochemistry.

Reviewer 2 Report

First of all I found the topic very interesting, when I accepted this review without knowing that this is a clinical study. The fact is that I’m more familiar in preclinical field (animal studies), therefore it is a little bit hard for me to evaluate the work.

I have found some smaller formal and methodological mistakes that should be corrected:

1.       In line 96: 55 patients are mentioned instead of 56 as it is in table 1

2.       The patients are divided into subgroups in the different analysis but the size of the subgroups (e.g. inflammatory – sever pain, or non inflammatory no pain or) (n) is missing from text, figures or legends. The n should be written in the legends at least.

3.       Also similar problem that it is not written how were those 33 patient chosen for double (proximal and distal) sampling. How many had inflammatory or non inflammatory disease? How many of those patient have no, mild or sever pain?

4.       In figure legend 1 Mann-Whitney test is mentioned with double cross (#) symbol, but on the figure I was not able to find this symbol. Had this test negative result on any parameter?

5.       In chapter 2.3.3 please write in the title the abbreviation in (IENFD).

As I mentioned for me it was really hard to understand the logical an mechanistic connection between results. I would suggest to prepare a figure for the discussion, and show the possible connections between the seven analyzed factors (TLR4, TRPV1, TNFalpha, SRTI1 and the miRNAs). If somebody is not deeply familiar with those pathways such a figure can help a lot to understand the situation. Maybe I’ve misunderstood something, but for example in the introduction (line 63-64) it is written that those miRNAs can inhibit the NFkB pathway, but in discussion (line 271-272) it is written, that miR-155-5p and 132-3p silence SRT1, however SRT1 is a inhibitor of NFkB (line 53). Please make a discussion somehow more understandable.

Reviewer 3 Report

An elegant study - nicely written up on an important subject. Few minor comments - perhaps a bit more discussion of existing Standard of Care and how that might be predicted to be less effective would be good. Also implications for future R&D and ultimately specific treatment especially for idiopathic PNP

Round 2

Reviewer 1 Report

Thanks for completing the corrections and adding clarifications in Figure 6.